# Environmental Exposure to Endocrine Disrupting Chemicals Influences Genomic Imprinting, Growth, and Metabolism

**DOI:** 10.3390/genes12081153

**Published:** 2021-07-28

**Authors:** Nicole Robles-Matos, Tre Artis, Rebecca A. Simmons, Marisa S. Bartolomei

**Affiliations:** 1Epigenetics Institute, Center of Excellence in Environmental Toxicology, Department of Cell and Developmental Biology, Perelman School of Medicine, University of Pennsylvania, 9-122 Smilow Center for Translational Research, Philadelphia, PA 19104, USA; nroble@pennmedicine.upenn.edu; 2Division of Hematology/Oncology, Boston Children’s Hospital, Harvard Medical School, Boston, MA 02115, USA; artis@g.harvard.edu; 3Center of Excellence in Environmental Toxicology, Department of Pediatrics, Children’s Hospital of Philadelphia, Perelman School of Medicine, University of Pennsylvania, 1308 Biomedical Research Building II/III, Philadelphia, PA 19104, USA; rsimmons@pennmedicine.upenn.edu

**Keywords:** endocrine disrupting chemicals, phthalates, bisphenol A, pesticides, DOHaD hypothesis, epigenetics, DNA methylation, genomic imprinting, growth, metabolism

## Abstract

Genomic imprinting is an epigenetic mechanism that results in monoallelic, parent-of-origin-specific expression of a small number of genes. Imprinted genes play a crucial role in mammalian development as their dysregulation result in an increased risk of human diseases. DNA methylation, which undergoes dynamic changes early in development, is one of the epigenetic marks regulating imprinted gene expression patterns during early development. Thus, environmental insults, including endocrine disrupting chemicals during critical periods of fetal development, can alter DNA methylation patterns, leading to inappropriate developmental gene expression and disease risk. Here, we summarize the current literature on the impacts of in utero exposure to endocrine disrupting chemicals on genomic imprinting and metabolism in humans and rodents. We evaluate how early-life environmental exposures are a potential risk factor for adult metabolic diseases. We also introduce our mouse model of phthalate exposure. Finally, we describe the potential of genomic imprinting to serve as an environmental sensor during early development and as a novel biomarker for postnatal health outcomes.

## 1. Introduction

Endocrine Disrupting Chemicals (EDCs) are natural or man-made chemicals capable of disrupting the regulation of the endocrine system in humans and animals [1]. EDCs can interfere with any aspect of endogenous hormonal action, including biosynthesis, metabolism, transport, elimination, or receptor binding of endogenous hormones [2]. This endocrine disruption leads to an imbalance in the maintenance of critical cellular homeostasis, thus ultimately increasing the risk of adverse health effects (i.e., metabolic syndrome, cancer, and abnormal behavior) [3,4]. Given that endogenous hormones regulate the physiology of target metabolic tissues, recent studies have demonstrated the effects of EDCs on sensitive metabolic processes [5,6,7]. Humans are exposed to EDCs from multiple sources, including plastic food containers, personal care products, medical devices, agricultural pesticides, and thermal receipts. Consistent with the ubiquitous presence of these chemicals in our environment, EDCs have been detected in human serum, urine, tissues, amniotic fluid, and breast milk [8,9,10]. 

Human exposure to EDCs starts as early as in the mother’s womb, where these chemicals have been demonstrated to cross the placenta and reach the fetus [11]. Developing fetuses and neonates are particularly vulnerable to EDC exposure because, at these developmental stages, the enzymes involved in the xenobiotic biotransformation and elimination of these chemicals are not completely functional [12]. As a result, excess accumulation of these chemicals causes detrimental effects on target organs and developing tissues (i.e., placenta, pancreas, developing gonads and the brain) [13]. In addition, early-life environmental exposures coincide with extensive epigenetic reprogramming that occurs during early embryogenesis and germ cell specification [14]. In the developing fetus, EDCs can modify the maintenance, reestablishment, and erasure of epigenetic marks, ultimately leading to increased susceptibility to adult diseases [15,16]. 

The Developmental Origins of Health and Disease (DOHaD) hypothesis, first proposed by David Barker and colleagues, suggests that environmental insults in critical periods of fetal development predispose offspring to diseases later in life (Figure 1) [17]. Multiple epidemiological and clinical studies show a strong correlation between low birth weight and a higher risk of cardiovascular and metabolic diseases manifested during adulthood [18]. Thus, environmental exposures, including EDCs, have become an emerging public health concern. Several animal models and human birth cohort studies have suggested that exposure to EDCs during early development can alter fetal growth and metabolism and subsequently disturb processes that promote metabolic disorders manifested during adulthood [7,19,20]. In particular, these chemicals have been associated with an increased risk of obesity, type 2 diabetes mellitus, and metabolic syndrome [20,21,22,23]. Although the mechanisms behind these associations remain unclear, epigenetic dysregulation has been proposed to have a role in gene-environment interactions and disease risk. Several studies have demonstrated that environmental perturbations, including assisted reproductive technologies (ART), prenatal famine, and EDCs, are associated with altered global and/or gene-specific DNA methylation patterns [24,25,26]. DNA methylation, the addition of a methyl group to the 5th carbon position in cytosines, is a widely studied epigenetic modification that regulates genes with critical roles in multiple biological processes [27]. This epigenetic modification has been associated with gene silencing at regulatory regions. Altered patterns of DNA methylation induce abnormal developmental gene expression resulting in abnormal phenotypic consequences [28]. Finally, DNA methylation is an important regulator of a subset of genes critical for fetal and placental development, called imprinted genes [29]. 

Genomic imprinting in mammals results in the monoallelic, parent-of-origin-specific expression of genes that are typically regulated predominantly by differentially methylated imprinting control regions (ICRs) [30,31]. DNA methylation at ICRs is established in the gametes, which in turn determines the allelic expression pattern of imprinted genes in the offspring [32]. Moreover, the ICRs of imprinted genes escape the genome-wide reprogramming that occurs during preimplantation, at least in part through the recognition of DNA methylation by specific Krüppel-associated box domain zinc finger proteins (KRAB-ZFPs) [33]. Given the susceptibility of DNA methylation marks to environmental insults, imprinted genes represent a robust model for EDC-induced epigenetic changes in disease as their regulation is dependent on epigenetic mechanisms [29,34]. In addition, imprinted genes are highly expressed in fetal and placental tissues to confer proper development of the conceptus and postnatal growth. Importantly, alterations in specific imprinted genes have been associated with cancer and developmental imprinting disorders such as Angelman Syndrome (AS) and Beckwith-Wiedemann Syndrome (BWS) [35]. Multiple studies have suggested that environmental exposure to EDCs during the earliest stages of fetal development may disrupt epigenetic reprogramming events and alter developmental genomic imprinting, ultimately leading to increased risk of adult diseases [36,37,38]. Thus, genomic imprinting may be one mechanism explaining how the genome and environment interact and how the environment influences the developmental origins of adult diseases in the context of chemical exposures. 

The reviewed animal and clinical studies summarize the prevailing thoughts regarding how early-life genomic imprinting alterations may be associated with adverse fetal growth trajectories and metabolic diseases during adulthood. We will discuss the value of using genomic imprinting profiling as a marker for environmental exposures to toxic chemicals and a robust epigenetic model to understand the fetal origins of adult diseases. Finally, we will discuss the remaining challenges in the field of environmental epigenetics and public health implications. These studies and future work will guide the scientific community in the prevention of chronic diseases associated with chemical exposures in vulnerable communities.

## 2. Genomic Imprinting: An Epigenetic Target of the Environment 

In 1976, the Dutch Hunger Winter study provided some of the first epidemiological evidence for a potential interaction between the environment and the epigenome [39]. The famine took place when the Germans occupied The Netherlands and rationed food such that the Dutch population, including pregnant women, received 400–800 calories per day [40]. One of the key findings was that the children of pregnant women exposed to famine early in gestation were more susceptible to diabetes, obesity, and other chronic metabolic diseases during adulthood [41,42,43]. The emergence of these metabolic diseases has been correlated with persistent environment-induced epigenetic differences, including alterations in DNA methylation patterns and genomic imprinting [39]. These findings suggest that environmental perturbations during intrauterine development increase susceptibility to epigenomic alterations [44,45]. 

The *Igf2* (insulin-like growth factor 2) and *H19* genes are imprinted in mice and humans. *Igf2* is a paternally-expressed growth-promoting factor, while *H19* encodes a non-coding RNA that is transcribed from the maternal allele [46]. *Igf2* is critical to fetal development because it promotes normal fetal and placental growth. *Igf2* imprinting is maintained via the *Igf2* ICR [39]. The misregulation of *Igf2* gene expression and epigenetic regulation have profound growth phenotypic consequences. For example, animal studies have shown that biallelic expression of *Igf2* results in overgrowth, whereas deletion of the *Igf2* gene results in growth deficiency [47,48,49]. Given the involvement of imprinted genes in epigenetic inheritance and fetal growth trajectories in response to environmental stimuli, it is possible that alterations in genomic imprinting may link abnormal fetal and postnatal growth to disease susceptibility later in life driven by maternal exposures. This has been supported by the Dutch Hunger Winter study, which showed that individuals exposed to famine in utero had hypomethylation at the *Igf2* DMR, six decades after initial exposure [39]. However, the changes in DNA methylation were small and have yet to be replicated in an independent cohort. Despite this, these studies provide a plausible mechanism linking early-life environmental exposures to the later development of disease. Thus, in this review, we evaluate genomic imprinting as a potential sensor of the environment driving fetal growth trajectories. 

## 3. EDCs Impact on Genomic Imprinting

As stated above, humans are ubiquitously exposed to EDCs by multiple routes [50]. EDCs have a number of different actions, but a key characteristic of these chemicals is that they exhibit non-monotonic dose responses [3], meaning that low-dose effects, which represent the range of human exposure, cannot be predicted by the effects observed in high doses used in toxicological studies. Given this characteristic, low doses of EDCs can cause adverse health effects if exposure happens during critical developmental periods [28]. This hallmark of EDCs represents a challenge to identify the possible mechanisms by which these chemicals act. Many xenobiotics, including EDCs, exert their effects through estrogenic and/or other hormonal properties [51]. However, their potential to modify epigenetic reprogramming events (i.e., genomic imprinting) remains largely unexplored. Finally, the epigenome is more vulnerable to environmental exposures during early development because, during this developmental window, dynamic changes in DNA methylation are required for normal tissue development [27]. Table 1 and Table 2 summarize recent animal and human studies, respectively, exploring the influences of EDC exposures on imprinted genes during vulnerable developmental periods including periconceptional, gestational, and early postnatal development.

### 3.1. Bisphenol A

Bisphenol A (BPA) is a synthetic chemical used in the production of polycarbonated plastics and epoxy resins, thus present in many products including thermal receipts, plastic bottles, food containers, and metal-based food cans. One major route of BPA exposure is through ingestion because this chemical is not covalently attached to the plastic and can leach into the food supply with changes in temperature and pH [52]. The ubiquitous presence of BPA in the environment raises public health concerns because of its ability to bind to nuclear estrogen receptors, thereby potentially altering the effects of estrogen [53]. In addition to hormone receptor binding, BPA may exert physiological and molecular changes on endocrine organs by altering the epigenome [28]. BPA was the first environmental contaminant screened to examine the adverse effects of EDC exposure on the fetal epigenome using the viable yellow agouti (*A^vy^*) mouse model [54,55]. This mouse model is used to correlate coat color variation with changes in epigenetic marks established during early development. Dolinoy et al. showed that maternal exposure to BPA shifted the coat color of *A^vy^/a* offspring toward yellow by inducing hypomethylation within the *A^vy^* intracisternal A particle (IAP) retrotransposon [36,54,55]. Additionally, they showed that BPA-induced hypomethylation was counteracted by maternal dietary supplementation with methyl donors (i.e., folic acid) [54]. These studies suggested that maternal exposure to BPA can influence offspring phenotype by altering the fetal epigenome. One limitation of using the *A^vy^* mice, which is approximately 93% C57BL/6J strain, is that *A^vy^* and C57BL/6J are not identical and cannot be considered as such [56,57,58]. Additionally, the observations made in these mice may not be extrapolated to genetically diverse human populations and outbred species [58,59]. 

The physiological and molecular effects associated with EDC exposures also depend on the timing and duration of the exposure [60,61]. During the development and differentiation of mouse germ cells, the genome undergoes extensive and dynamic epigenetic reprogramming. Erasure of imprinting marks in mouse primordial germ cells (PGCs) occurs around embryonic (E) day 10.5–11.5 while imprinting establishment occurs during gametogenesis [62,63]. To evaluate the effects of BPA gestational exposure on mouse fetal germ cells, Zhang et al. interrogated the DNA methylation status of imprinted genes in E12.5 PGCs [64]. They showed that DNA methylation of imprinted genes *Igf2r*, *Peg3* and *H19* had an average decrease of 12–30% in fetal mouse germ cells with greater effects in the BPA groups treated with relatively high doses. In contrast, Iqbal et al. found no changes in allele-specific DNA methylation of imprinted genes in either female or male E13.5 germ cells following BPA in utero exposure [65]. However, their allele-specific expression analyses showed sex-specific effects of loss of imprinting (LOI) at the imprinted gene *Meg3* in BPA-exposed female germ cells (FGCs) [65]. 

Given that de novo establishment of oocyte DNA methylation takes place after birth in growing oocytes of juvenile females, Chao et al. tested the potential effects of BPA on DNA methylation of imprinted genes during mouse oocyte growth (~PND5-PND20) [32,66]. This study reported decreased DNA methylation at the differentially methylated regions (DMRs) of imprinted genes *Peg3* (83% control vs. 12% in high dose BPA) and *Igf2r* (90% control vs. 40% high dose BPA) of BPA exposed growing oocytes (PND15, PND21) [66]. These changes during postnatal development correlated with what Zhang et al. observed in E12.5 PGCs. However, the variable results between these three studies could be attributed, in part, to differences in experimental design. Iqbal et al. used a shorter exposure window while Chao et al. focused on postnatal exposure as opposed to in utero exposure used in the other studies. Additionally, this study used different routes of exposure and doses. Although these studies used low and biologically relevant doses of BPA, one limitation is that it is difficult to translate dosages from animal experiments to human exposures. Additionally, humans are exposed to mixtures of EDCs, causing additive or reductive actions that are not comparable to animal exposures. Finally, BPA and other EDCs elicit non-monotonic dose responses, meaning that the mode of action of BPA at each dose could yield different molecular and/or physiological outcomes [3]. 

Recently, the placenta has been considered a major driver of disease risk in offspring [67,68]. This organ forms the gestational interface between the fetus and the mother and controls exchange of nutrients, oxygen, and waste products. The placenta has endocrine functions and acts as a barrier to minimize exposure of the fetus to maternal xenobiotics, hormones, and pathogens. However, if placental function is impaired by environmental perturbations including EDCs, then fetal development may be compromised. Kang et al. tested whether BPA exposure during mid-gestation causes epigenetic perturbations at imprinted genes of E13.5 mouse placentas [29]. This study showed that in utero BPA exposure reduced the expression of the imprinted gene *Rtl1as* in the placenta and *Slc22a18* in the whole embryo. In contrast, the experimental design of Susiarjo et al. not only took into consideration maternal BPA exposure during early stages of embryonic development but also late stages of oocyte development (preconception). This latter study demonstrated that maternal BPA exposure was associated with LOI at imprinted genes *Ascl2* (placenta-specific imprinting), *Kcnq1ot1*, and *Snrpn* of E9.5 placentas. To correlate allelic-specific expression with allelic-specific DNA methylation, they showed that BPA exposure significantly altered DNA methylation only at the *Snrpn* ICR of E9.5 placentas. A recent study evaluated whether Bisphenol S (BPS), a BPA substitute, influences mouse placental development and imprinted gene expression at mid-gestation, similar to the study of Susiarjo et al. [69]. They reported reduced mRNA levels of *Ascl2* in both BPS and BPA treated groups, but the differences were significant only in the BPA-treated group [70]. These inconsistent findings can be attributed to the use of different mouse strains, doses, and routes of administration. Both Susiarjo et al. and Mao et al. used oral feed as the route of administration, which is more comparable to human exposure, as opposed to oral gavage used by Kang et al. [25,29,69]. With respect to global DNA methylation, only Susiarjo et al. reported reduced placental DNA methylation following early life exposure to BPA [25]. Given that the placenta responds to the environment in a sex-specific manner, one limitation of these studies is that results were not stratified by sex. Thus, future studies should include sex as a variable. Additionally, as new technologies become available, postnatal physiological outcomes may be better correlated with placental epigenetic changes, which may allow the use of the placenta as a biomarker of offspring health. Even with these caveats, these studies demonstrated that EDCs can modify the epigenome of the placenta, which may contribute to poor placental development and function and consequently fetal health outcomes. 

To evaluate if genomic imprinting and epigenetic changes persist postnatally, several studies examined BPA exposure from gestation through lactation, and postnatal offspring tissues were assayed. Drobná et al. conducted transcriptomic analyses to investigate which genes in the brain are changed transgenerationally by gestational exposure to BPA in the F3 generation [70]. Interestingly, in utero BPA exposure in F0 dams resulted in unexposed F3 male offspring with significantly higher mRNA expression of the imprinted gene *Meg3* at PND28 [70]. To correlate expression changes with DNA methylation levels, pyrosequencing was used to probe the DMRs of *Meg3*. At the *Meg3* promoter, 3 CpG sites exhibited decreased CpG methylation in F3 male brains, but the changes were quite small [70]. Because the differences in *Meg3* transcript levels were inconsistent with changes in DNA methylation (hypermethylation), altered DNA methylation is unlikely to fully explain the differences in *Meg3* gene expression observed in the F3 generation. To examine whether the effects of BPA exposure and age alter epigenetic drift, Kochmanski et al. developed longitudinal measures via targeted assays of locus-specific methylation in paired mouse tails (3 weeks/10-month-old) [71]. They reported that across age, there was a trend towards reduced DNA methylation levels at two repetitive elements (LINE-1 and IAP) and one imprinted gene (*H19*) [71]. Malloy et al. used the same exposure model as Kochmanski et al. to assay adult brains (10 months) [72]. The reported data demonstrated that perinatal BPA exposure is associated with altered expression of the imprinted gene *Kcnq1* in the adult mouse brain, but no changes in DNA methylation were observed [72]. Thus, further studies with larger sample sizes and multiple timepoints are necessary to understand the mechanisms and biological pathways driving these changes and to determine the implications of these changes for brain development. 

### 3.2. Phthalates

Phthalates are a group of man-made chemicals known as plasticizers due to their use in making plastics more durable and flexible [73]. Historically, some phthalates have been used as solvents and fixatives in fragrances and other materials. Other phthalates are used in the production of polyvinyl chloride (PVC) materials. Humans are exposed to phthalates through toys, food packaging, medical devices, detergents, personal care products such as nail polish, lotions, and perfumes, as well as occupational exposures from PVC production [73,74]. For this reason, there is a ubiquitous presence of phthalates in our environment. Consistently, their metabolites have been detected in >75% of the U.S. population [75]. The most common phthalate, Di-2-ethyhexyl-phthalate (DEHP), is an exogenous chemical with the capability to impair testosterone synthesis, resulting in anti-androgen effects [76]. Given the anti-androgenic properties associated with DEHP and its metabolites, initial environmental and public health studies were focused on the effects of DEHP on male reproduction. Both human and animal studies reported that environmental exposure to DEHP was associated with male infertility, poor sperm count and quality, and testicular dysgenesis syndrome (TDS) [77,78,79]. However, recent animal and birth cohort studies suggest that DEHP exposure in critical periods of development may also influence epigenetic reprogramming events including genomic imprinting [29,65,80,81,82]. 

Iqbal et al. evaluated the effects of DEHP on global epigenetic reprogramming and imprint resetting in the male germline after in utero exposure [65]. The imprinted gene *H19* was affected by DEHP in fetal germ cells, where LOI was observed. To test whether DEHP exposure perturbs the imprinting process in prospermatogonia, this group tested allele-specific DNA methylation patterns at DMRs in the soma of F2 offspring [65]. Analysis of allele-specific methylation at paternal DMRs showed a significant increase (>5%) in the *Rasgrf1* DMR of F2 head and heart exposed to DEHP through F0 dams [65]. Notably, the dose of DEHP use in this study (750 mg/kg/d) is extremely high compared with a reference dose of 20 μg/kg/d for the general population. Given the non-monotonic nature of EDCs, the outcomes associated with these toxic and high doses may not be representative of changes occurring at lower biologically relevant doses. To overcome this challenge, Li et al. used low dose DEHP (40 μg/kg/d) to assess its effects on DNA methylation of imprinted genes in germ cells from fetal and adult mice [81]. They showed that in utero exposure to DEHP significantly reduced the DNA methylation at *Igf2r* and *Peg3* DMRs in both female and male PGCs. Additionally, these modifications in DNA methylation were present in the F2 offspring [81]. Thus, DEHP not only affected fetal mouse germ cells and growing oocytes, but also the F2 offspring’s oocytes.

The scientific literature on the effects of DEHP on the mouse placenta is limited. Multiple studies in pregnant rats reported associations between maternal exposure to DEHP and reduced embryo implantation, increased resorptions, and decreased fetal weight [83,84,85]. However, no detailed information is currently available about the effects of DEHP on placental function and epigenetic changes during pregnancy. The work of Zong et al. evaluated the effects of DEHP on placenta and embryo development [80]. Although the experiments were not focused on genomic imprinting, they did indeed find reduced mRNA levels of *Ascl2* in E12.5 placentas exposed to the highest doses of DEHP [80]. One limitation of this study is that the doses used were very high and considered “toxic” doses. Thus, future experiments should include lower doses of DEHP to compare with the higher doses.

### 3.3. Pesticides 

Vinclozolin (VZ) is a dicarboximide fungicide used to treat fruits and vegetables such as grapes, lettuce, and beans and its use was banned in the U.S. in 2006 [86]. VZ or its active metabolites (M1 and M2) inhibit the binding of androgens to their receptors, which results in anti-androgen effects [87]. VZ exposure in utero has been associated with reduced prostate weight in adult rats and decreased anogenital distance of male offspring at birth. VZ exposure during the gestational period of sex determination caused longer urethras in female offspring, hypospadias in male offspring, and increased expression of progesterone receptors in both sexes [88]. Furthermore, VZ altered spermatogenesis by inducing sperm head abnormalities, decreased sperm count and motility, and apoptosis in seminiferous tubules of germ cells. Surprisingly, these effects in spermatogenesis and male genital tract were found to be transmitted transgenerationally from F1 to F4 [89,90]. Given the transgenerational effects of VZ in male reproduction, recent studies are investigating possible epigenetic effects in the male germline following in utero VZ exposure. 

To address the involvement of epigenetics in the reported male reproductive outcomes in adulthood, Stouder et al. evaluated the adverse effects of VZ in utero exposure on imprinted genes. They found that sperm of VZ-exposed adults (PND56) exhibited decreased CpG methylation at *H19* and *Gtl2*, and increased methylation at *Peg1*, *Snrpn*, and *Peg3* [91]. This study proposed that the effects induced by VZ in the male reproductive system could be, in part, due to imprinting defects in sperm. Pietryk et al. took a similar experimental approach, but they examined adult sperm at a later stage (PND84) [92]. This work used genetic mouse lines carrying mutations at the *Igf2/H19* ICR to determine the effects of different genetic sequences on phenotypic and epigenetic outcomes following VZ exposure during fetal development [92]. First generation offspring from VZ-treated 8nrCG mutant dams (mutation of 8 CpGs outside of CTCF binding sites) exhibited a small reduction in sperm *Igf2/H19* ICR methylation [92]. These studies show that comparing EDC effects on multiple genetic lineages will benefit risk assessment. Although both studies used the same VZ doses and dermal injection as the route of administration, Pietryk et al. were unable to replicate the findings of Stouder et al. on sperm methylation changes at the *Igf2/H19* ICR, which could be explained by different mouse strains and phenotypic timepoints used in the studies and/or other environmental factors (i.e., diet). Future studies should evaluate the mouse strain as a variable and perform genome-wide analyses to determine if other epigenetic modifications play a role in these molecular and phenotypic changes induced by VZ. 

### 3.4. Dioxin

The EDC 2,3,7,8-Tetracholodibenzeno-*p*-dioxin (TCDD) belongs to the dioxin/dioxin-family of environmental toxicants [93]. TCDD is introduced to the environment as a by-product of industrial processes such as incineration and burning of fossil fuels but can also result from natural processes such as volcanic eruptions and forest fires [94]. Among human and animal populations, ingestion of TCDD-contaminated food is the primary source of dioxin exposure [95,96,97]. Once TCDD enters the body, it is chemically stable and not readily metabolized in most species [98]. In humans, TCDD has a half-life of 8 years and is highly resistant to either biological or chemical degradation [99]. Thus, this dioxin exhibits a significant degree of environmental persistence and bioaccumulation. The first toxic effects of TCDD were publicly available after the Vietnam War and the Seveso accident. During the Vietnam War, TCDD-containing defoliants were reported to be the cause of a drastic increase in pregnancy loss and birth defects [100,101]. In 1976, a toxic cloud of TCDD was released into the environment as a consequence of an accident in a chemical plant in Seveso [102]. Here, adult men exposed to TCDD in utero or as a child exhibited low sperm counts and decreased motility [103,104]. Although TCDD can affect the reproductive system through its disruption on steroid receptor levels, steroid metabolism, and transport [105,106], it is of great interest to better understand its effect on epigenetics and genomic imprinting. 

Wu et al. explored the possible adverse effects of TCDD on imprinted genes during the earliest stages of embryonic development [107]. The results showed that early-life exposure to TCDD increased (>20%) methylation of the *Igf2/H19* ICR and decreased expression of *Igf2* and *H19* in the whole embryo [107]. To follow up on these findings, Somm et al. evaluated the deleterious effects of TCDD on imprinted genes of adult male offspring (PND56) [108]. They reported 0.5–1.5-fold increased mRNA levels of imprinted genes *Snrpn*, *Peg3* and *Igf2r* (sperm), and 0.5-fold decreased expression of *Igf2r* (muscle) [108]. They correlated decreased expression of *Igf2r* in muscle with increased number of methylated CpGs in *Igf2r* [108]. 

**Table 1 genes-12-01153-t001:** Animal Studies.

EDC(s)	Dose(s)	Route of Exposure	Rodent Strain	F1 Exposure Window	F1 Age(s) at Endpoint	Tissue(s) Assayed	Genomic Imprinting Change(s)	Reference
BPA	0.5, 20, or 50 μg/kg/d	Oral (pipette)	B6	E11—birth	PND0, PND4, PND28	F3 generation brain regions	Gene expression: BPA exposed males had significantly higher mRNA expression of *Meg3* than females DNA methylation: no changes in the IG-DMR methylation status. At the *Meg3* promoter, 3 CpG sites were hypermethylated in male brains	[70]
BPA, DEHP, VZ	BPA: 0.2 mg/kg/d DEHP: 750 mg/kg/d VZ: 100 mg/kg/d	Oral (gavage)	JF1 × OG2 hybrid mice	E8.5–E12.5	E13.5	Whole embryo Lung and placenta Yolk sac	Relaxation of imprinted gene expression of: *Slc22a18* (BPA) *Rtl1as* (BPA) *Rtl1* (DEHP)	[29]
BPA	Low: 10 μg/kg/d High: 10 mg/kg/d	Oral (feed)	B6 × C7 hybrid mice	E0–E9.5 E0–E12.5	E9.5, E12.5	Whole embryo, Placenta	Allele-specific expression: LOI at imprinted genes *Ascl2*, *Kcnq1ot1*, *Snrpn*, *Igf2* Gene expression: total mRNA expression increased for imprinted genes *Snrpn*, *Igf2*, *Kcnq1ot1*, decreased for *Cdkn1c* and *Ube3a* DNA methylation: reduced at *Snrpn* ICR, increased at *Igf2* DMR1, reduced global DNA methylation	[25]
BPA, DEHP, VZ	BPA: 0.2 mg/kg/d DEHP: 750 mg/kg/d VZ: 100 mg/kg/d	Oral (gavage)	JF1 × OG2 hybrid mice JF1 × 129S1	E8.5–13.5 E12.5–E16.5	E13.5 E13.5	Female germ cells (FGCs) Male germ cells (MGCs)	Allele-specific expression: FGCs: LOI at imprinted genes *Meg3* (BPA) and *H19* (DEHP) MGCs: LOI at the imprinted gene *Meg3* (VZ) Allele-specific DNA methylation: LOI at the *IG-DMR* in liver and head (VZ), *Rasgrf1* DMR in head and heart (DEHP), *Rasgrf1* DMR in head (VZ)	[65]
BPA	0, 40, 80, 160 μg/kg/d	Oral (gavage)	CD-1 mice	E0.5–E12.5	E12.5	Fetal mouse germ cells (Primordial Germ Cells)	DNA methylation: decreased in imprinted genes *Igf2r*, *Peg3* and *H19* DMRs	[64]
BPA	0, 20, 40 μg/kg/d	Dermal (hypodermical injection)	CD-1 mice	PND7–PND15 PND5–PND20 (injection every 5 days)	PND15 PND21	Mouse oocytes	DNA methylation: decreased at the DMRs of the imprinted genes *Peg3* and *Igf2r*	[66]
BPA	50 μg/kg/d	Oral (feed)	B6 × *A^vy^*/*a* (viable yellow agouti)	E0–PND21	10 months	Brain cortex and midbrain	Gene expression: higher gene expression of *Kcnq1* DNA methylation: no alterations in 5mC levels	[72]
BPA	50 μg/kg/d	Oral (feed)	*a/a* × *A^vy^*/*a*	E0–PND21	10 months	Tail	DNA methylation: decreased at *H19*, *Igf2*, IAP, and LINE-1	[109]
BPA, BPS	200 μg/kg/d	Oral (feed)	B6	E0–E12.5	E12.5	Placenta	Gene expression: reduced mRNA levels of *Ascl2*	[69]
VZ	50 mg/kg/d	Dermal (injection)	B6 × C7 hybrid mice	E9.5–E18.5	PND84	Sperm	DNA methylation: reduced at *H19/Igf2* ICR	[92]
VZ	50 mg/kg/d	Dermal (i.p. injection)	FVB/N mice	E10–E18	PND56	Sperm	DNA methylation: number of methylated CpGs decreased in *H19* and *Gtl2*, and increased in *Peg1*, *Snrpn* and *Peg3*	[110]
TCDD (dioxin)	2, 10 ng/kg/d	Dermal (i.p. injection)	FVB/N mice	E9–E19	PND56	Sperm, liver, muscle	Gene expression: increased mRNA levels of imprinted genes *Snrpn*, *Peg3* and *Igf2r* (sperm), decreased expression of *Igf2r* (muscle) DNA methylation: increased the number of methylated CpGs in *Igf2r* (muscle)	[108]
DEHP	40 μg/kg/d	Oral (gavage)	CD-1 mice	E0.5–E12.5	E12.5	Primordial germ cells	DNA Methylation: reduced at *Igf2r* and *Peg3* DMRs	[81]
DEHP	125, 250, 500 mg/kg/d	Oral (gavage)	CD-1 mice	E1–E9 E1–E13	E9, E13	Placenta	Gene expression: reduced mRNA levels of *Ascl2*	[80]

### 3.5. Human Studies 

The animal studies highlighted in the previous sections have shown associations between EDC exposure in utero and alteration of global and site-specific DNA methylation. However, extrapolating these findings to the human population is challenging in part because the animal studies investigated individual EDCs, while humans are exposed to a mixture of EDCs. Tindula et al. examined the association of prenatal phthalate exposure and imprinted gene DNA methylation profiles in cord blood of newborn children. In newborns of the Center for the Health Assessment of Mothers and Children of Salinas (CHAMACOS) study, prenatal exposure to several DEHP metabolites was positively associated with DNA methylation at the *MEG3* DMR [111]. Given that previous birth cohorts only investigated 2–3 imprinted genes, one of the strengths of this study was the evaluation of DNA methylation profiles of multiple imprinted genes in newborns prenatally exposed to phthalates [112,113]. A potential limitation of this study involves the applicability of these findings to other human populations, given that CHAMACOS participants were exclusively Mexican Americans. A case control study from Zhao et al. in third-trimester placentas from Chinese mother-newborn pairs found inverse associations between placental *Igf2* DNA methylation and maternal urinary phthalate metabolite concentrations [113]. A strength of this study was assessment of the placenta, which is a target tissue directly responsible for fetal growth and health. Some limitations of this study include relatively small sample size and analysis of only two growth-related imprinted genes. The human studies mentioned above did not show sex-specific effects on DNA methylation following prenatal EDC exposure. However, the Michigan Mother-Infant Pairs (MIIP) birth cohort reported an inverse correlation between *Igf2* methylation and urinary BPA concentrations in females only [114]. Future studies may incorporate longitudinal analyses and investigation of other epigenetic modifications to overcome the inconsistencies between gene expression and DNA methylation changes. 

**Table 2 genes-12-01153-t002:** Clinical Studies.

EDC(s)	Study Design	Study Population	Gestational Age at Sampling and Sampling Site	EDC Levels	Outcomes	Reference
Phthalates	CHAMACOS Longitudinal Birth Cohort	United States Mexican-American women and their newborn children *n* = 296	296 first and third trimester maternal urine and whole cord blood (148 girls, 148 boys)	Phthalates (μg/g creatinine): Σ DEHP: 60.9 MEHP: 3.9 MEOHP: 12.1 MEP: 214.2 MECPP: 26.7 MEHHP: 16.1	Positive association between pregnancy DEHP metabolites and HMW phthalates and methylation percent at *MEG3* DMR; negative associations between DEHP metabolites and *MEG3* expression. Lower average *MEG3* DMR methylation associated with low birth weight	[111]
Phthalates	Case-Control Study	China *n* = 220	Third trimester urine samples from 220 mother-newborn pairs and term placentas	Σ DEHP: 25.5 ng/mL, MBP: 25.7 ng/mL, MMP: 8.1 ng/mL, MEHP: 3.8 ng/mL, MEHHP: 10.8 ng/mL, MEOHP: 4.2 ng/mL	Inverse association between placental *Igf2* DNA methylation and maternal third trimester urinary phthalate metabolite concentrations. This association was stronger in the fetal growth restriction newborns.	[112]
BPA (*n* = 56), phthalates (*n* = 109)	Michigan Mother Infant-Pairs (MMIP) Birth Cohort	United States *n* = 116	First trimester (18–14 weeks pregnancy) Maternal urine samples and infant cord blood	BPA: 0.57 ng/mL (urine) BPA: 0.78 ng/mL (plasma) Sum DEHP Metabolites: 0.09 nMol (urine), 0.9 ng/mL (plasma)	Inverse correlation between *Igf2* methylation and urinary BPA concentration (females only).	[114]
Phthalates (MEP, MBP, MIBP, MCPP, MBzP, MECPP, MEHHP, MEHP, MEOHP, DEHP)	Second and third cohort of the ELEMENT longitudinal study	Mexico *n* = 1079 mothers *n* = 250 children	First, second and third trimester maternal spot urine (phthalate metabolites analysis) Children (9–17 years) whole blood (DNA methylation analysis)	Third trimester concentrations: MEP: 112.8 μg/L MBP: 57.27 μg/L MIBP: 2.12 μg/L MCPP: 1.13 μg/L MBzP: 4.30 μg/L MECPP: 31.75 μg/L MEHHP: 19.38 μg/L MEHP: 5.42 μg/L MEOHP: 11.89 Σ DEHP: 76.69	MBzP exposure increases *H19* DNA methylation, which is positively associated with increased adiposity in girls.	[115]
BPA	Congenital Anomaly Study cohort (mothers) Environment and Development of Children (EDC) prospective cohort (children)	Seoul, Korea *n* = 726 children [2-year old (*n* = 425) and 4-year old (*n* = 301)]	Second trimester maternal urine (*n* = 59 mothers) Whole blood (*n* = 59 children)	Urinary BPA: 1.34 μg/g creatinine	Increase *Igf2r* methylation levels in the high dose BPA group at age 2 years but not at the age 6. Positive association between BMI at 2 years and *Igf2r* DNA methylation (in girls but not in boys).	[116]

## 4. Early-Life Edc Exposure: A Potential Risk Factor for Adult Metabolic Diseases

The endocrine system is intimately involved in growth, weight, and metabolic processes through the production of hormones and growth factors that function through a series of tightly integrated signaling pathways [117]. Hormones regulate the physiology of pancreas, muscle, liver, and adipose tissue systems. EDCs are capable of disrupting hormonal regulation by mimicking or blocking normal endocrine functions, which can lead to metabolic diseases [5]. The current national increase in metabolic diseases correlates with substantial increases in chemical production and exposure in our environment [22]. Epidemiological and animal studies have demonstrated that perinatal environment plays a critical role in adult metabolic health [20]. Song et al. showed that perinatal BPA exposure in rats induced hyperglycemia, which contributes to insulin resistance in adult males [118]. Similarly, Garcia-Arevalo et al. reported that BPA exposure during pregnancy was associated with hyperglycemia, hyperinsulinemia, and insulin resistance in F1 adult offspring [119]. These metabolic changes mimic the effects found in high fat diet-fed mice not exposed to BPA. 

To provide novel insights into mechanisms driving multigenerational metabolic abnormalities, Susiarjo et al. investigated the maternal metabolic milieu and inheritance of DNA methylation across generations. They found that both F1 and F2 male offspring perinatally exposed to the highest dose BPA (10 mg/kg/d) were fatter and developed insulin resistance during adulthood [120]. In contrast, their islet perfusion analysis revealed that F1 and F2 male mice exposed to the lower dose BPA had impaired glucose-stimulated insulin secretion [120]. Also, both F1 and F2 embryos had increased DNA methylation at the *Igf2* DMR1 and elevated total *Igf2* mRNA expression. Thus, they were able to associate the observed multigenerational metabolic phenotypes in this BPA exposure model with persistent epigenetic changes. One limitation of this study includes the unknown role of estrogen receptors mediating the BPA-induced multigenerational alterations in metabolism and epigenetic changes. Using the same mouse model, Bansal et al. investigated potential mechanisms driving these metabolic changes in F1 and F2 male offspring [121]. They found that upper dose BPA induced islet inflammation and impaired mitochondrial function in F1 offspring that persisted into the next generation [121]. Additionally, they reported that the *Igf2* expression and DNA methylation alterations described in Susiarjo et al. persisted in F1 islets in adulthood and into the next generation. The mechanisms driving the transmission of these metabolic changes are unclear, but these studies suggested *Igf2* has a role through epigenetic dysregulation. 

Animal and human studies show that developmental exposure to phthalates is associated with an increased risk of metabolic syndrome. For example, Lin et al. used a rat model to reveal that gestational exposure to DEHP impairs the function of the F1 adult female endocrine pancreas and, in addition, causes insulin resistance in F1 adult males [122]. Additional animal studies have linked DEHP exposure with β-cell dysfunction in both sexes, while others report an increase in obesity only in males [123,124]. Human epidemiological studies correlated prenatal and postnatal DEHP exposure with childhood obesity and insulin resistance in adult males [125,126]. These sex-specific metabolic phenotypes observed in EDC-exposed offspring may arise from different abilities of males and females to adapt to environmental stimuli in utero, highlighting the potential for sexually dimorphic fetal programming and disease risk. 

Neier et al. explored perinatal exposures to DEHP, diisononylphthalate (DINP), and dibutyl phthalate (DBP) and how these mixtures impact metabolic outcomes compared with individual compounds [127]. They showed that perinatal exposure to DINP only increased body weight of males and females at PND21 and that use of mixtures did not exacerbate these results [127]. Also, females perinatally exposed to DEHP + DINP + DBP had an increased relative gonadal fat pad weight [127]. They also followed mice longitudinally (3 weeks vs. 10 months old) to explore long-term metabolic outcomes. This analysis revealed that perinatal DEHP exposure increased body fat and decreased lean mass in females, while perinatal DINP exposure impaired glucose tolerance in females [37]. However, in this longitudinal analysis, no persistent body weight changes were detected. A smaller sample size may contribute to the failure to detect robust changes in body weight. Other investigators have reported conflicting results regarding perinatal phthalate exposure effects on body weight, showing both increased and decreased body weights in adult rats and mice [128,129,130]. The inconsistencies among these studies might be explained by different mouse strains used, including outbred (CD-1) and inbred (B6, C3H/N) mice, and different end points of body weight measurements (PND21, 8 weeks, 12 weeks, and 10 months). 

Results from animal studies investigating the harmful effects of phthalates on glucose metabolism are controversial. Martinelli et al. found that Wistar rats fed a diet containing 2% DEHP had glucose intolerance [131]. In contrast, Feige et al. detected no changes in glucose metabolism in DEHP-exposed mice [132]. Kwack et al. investigated short-term metabolic effects of phthalate monoesters and diesters on rats [133], reporting that male rats exposed to DEHP, mono-ethyhexyl-phthalate (MEHP), and monobutyl-phthalate (MBP) exhibited higher glucose levels compared with control groups [133]. Our lab developed a mouse model to investigate the impacts of periconceptional DEHP exposure on the development of adult metabolic outcomes. In our exposure paradigm, virgin C57BL/6J (B6) females, 6–8 weeks old (designated F0 generation), were assigned to a low phytoestrogen control diet with or without the addition of DEHP. Animals were exposed to DEHP through the diet, which is representative of human exposure. In an initial dose response study, we used 0, 0.05, 1, 10, and 100 mg DEHP/kg bw/day. The F0 dams were exposed to DEHP-containing feed starting two weeks prior to mating until weaning. After weaning, F1 offspring were fed either a low-phytoestrogen control diet or a high calorie Western diet (WD) until 6 months of age (PND182). We found that in utero exposure of F0 dams to the highest dose of DEHP (100 mg/kg bw/day) resulted in fetal death by E17.5 and dams from two subsequent cohorts were unable to deliver live offspring (Figure 2A). Thus, we selected 0.05 mg/kg bw/day as the lower DEHP dose and 10 mg/kg bw/day as the upper DEHP dose for our subsequent studies, which are non-toxic and representative of human exposure levels. 

We first investigated if periconceptional DEHP exposure causes sexually dimorphic phenotypes in F1 adult offspring. We performed metabolic phenotyping in F1 adult offspring including growth trajectories, glucose tolerance tests at PND140, and percent body fat at the end of exposure (PND182). Lower (0.05 mg/kg bw/day) and upper dose (10 mg/kg bw/day) DEHP exposure had no effects on body weight, body fat content, and glucose tolerance (Figure 2 and Figure 3). Because initial phenotypes after DEHP exposure were unremarkable, we further challenged the animals to a high calorie Western Diet (WD) to exacerbate the metabolic effects. The F1 female offspring exposed in utero to lower dose DEHP and then fed a WD post-weaning, had a trend towards glucose intolerance (*p* = 0.0869) and increased body weight (*p* = 0.2491) and body fat (*p* = 0.3132). However, these changes did not reach statistical significance compared to the control groups (Figure 2B–F). Further, the WD challenge effects were observed in both control and DEHP treatment groups, suggesting that the changes were due to the WD and not to the periconceptional DEHP exposure. In contrast, F1 male offspring periconceptionally exposed to upper dose DEHP (10 mg/kg bw/day) and then fed a WD post-weaning showed trends towards glucose intolerance (*p* = 0.6713), and increased body weight and body fat (*p* < 0.05) (Figure 3A–E). Although these observations showed sex- and dose-specific trends, the changes were not significantly different from control groups, and all WD challenged offspring had similar effects attributed to the WD post-weaning and not to DEHP exposure in utero. Previous studies have also shown either variable or very modest effects of developmental DEHP exposure on metabolic parameters [37,127,131,134]. Why DEHP is more variable than other toxicant exposures is unclear. Although the published studies use different mouse strains, exposure routes and durations, these results demonstrate that DEHP causes subtle changes in glucose homeostasis and growth that are not as robust as other endocrine disruptors and larger studies as well as additional endpoints are necessary. 

## 5. Conclusions

The regulation of imprinted genes is critical for normal development of the fetus. Given that genomic imprinting is established during early development, environmental exposures that alter the maternal gestational milieu at this developmental window can result in an increased risk of abnormal metabolic outcomes in adulthood. Studies from humans and rodents highlighted in this review reveal the effects of EDC maternal exposure on genomic imprinting, growth, and glucose homeostasis of the offspring and subsequent generations. This literature indicates that imprinted genes are susceptible to environmental factors during specific developmental windows, suggesting that this epigenetic mechanism serves as an environmental sensor. It remains to be determined if these epigenetic changes are causal to an abnormal metabolic phenotype. However, given the importance of imprinted genes to metabolic function, it is likely that these two processes are linked. These studies and future work will guide the scientific community in the prevention of chronic diseases associated with chemical exposures in vulnerable communities. Prevention and intervention strategies with culturally appropriate messages for pregnant women should be implemented to promote an environment for optimal growth of the fetus.

It should also be noted that we have primarily focused on epigenetic perturbations of imprinted genes regulated by DNA methylation-dependent (canonical) imprinting. More recent studies have identified a class of genes regulated by H3K27me3-mediated non-canonical imprinting [135]. These genes are critical for placental development, maintaining imprinted gene expression in the extraembryonic lineage. Thus, we anticipate that future studies will also include these genes, as well as effects of the environment on chromatin modification. 

## 6. Recommendations for Future Research on Edcs and Environmental Epigenetics

To overcome the remaining challenges in the EDC developmental exposure research field, we suggest the following: Windows of EDC susceptibility: Incorporate periconceptional (preconception) and/or early postnatal (lactation) developmental windows in future environmental epigenetic studies.Toxic doses vs. physiological doses: Focus on the physiological and molecular effects of EDCs at low doses.Individual compounds vs. mixtures: Integrate EDC mixtures in future investigations to determine if these mixtures have additive or diminished effects on fetal growth and metabolism.Dose administration and rodent strains: Establish an agreement between multiple labs to use similar dose of administration and rodent strains to compare the reproducibility of EDC effects. Use dose administration in the animal feed because is more relevant to the main route of human exposure (ingestion).Sex-specificity: Report both molecular and physiological endpoints by sex of the fetus.Epigenetic modifications: Include other epigenetic marks (i.e., histone modifications) in the molecular analyses of EDC exposure models.Mechanisms: Future work should focus on the mechanism(s) driving ICR dysregulation after EDC exposure.

## 7. Methods 

### 7.1. Experimental Animals

Six-week-old virgin C57BL/6J females were purchased from The Jackson Laboratory and assigned to the following diets: (1) modified AIN 93G diet (TD 95092 with 7% corn oil substituted for 7% soybean oil; Harlan Teklad, Madison, WI, USA) as “control”; (2) modified AIN 93G diet supplemented with 250 ppb DEHP (TD 140177; Harlan Teklad, Madison, WI, USA) as “lower dose”; or (3) modified AIN 93G supplemented with 50 ppm DEHP (TD 140175; Harlan Teklad) as “upper dose”. Teklad Diets (Harlan Laboratories Inc) provided all ingredients except DEHP (>99% in purity; Sigma-Aldrich, St. Louis, MO, USA). Based on the average body weight (bw) of an adult mouse (25 g) and daily food consumption (5 g), we estimated exposure levels per mouse to be 50 μg/kg bw/d (lower dose) and 10 mg/kg bw/d (upper dose). These doses were selected based on dose response analyses demonstrating the effects of DEHP exposure on RNA expression at imprinted loci. After 2 weeks of treatment, females were time-mated to C57BL/6J males, and the day a plug was detected was assigned as embryonic day (E) 0.5. Pregnant females were designated as the “F0”. For dose response analyses, F0 females were euthanized at E17.5 and embryos were collected. For the rest of the experiments, F0 females were allowed to deliver their offspring until weaning; the offspring were designated as the “F1”. At postnatal day (PND) 21, all F1 mice were weaned on either the TD 95092 (control) diet or TD.180081 (45% kcal fat diet), so these mice were exposed to DEHP during gestation and lactation periods only. Body weight was recorded weekly starting at PND 1 until PND182 and at the time Dual Energy X-ray Absorptiometry (DEXA) was performed to measure body fat. 

### 7.2. Glucose Tolerance Tests (GTTs)

At PND140 (adult F1 offspring) mice were fasted for 6 hours and subsequently injected with 2 g/kg bw of glucose intraperitoneally. At 0, 15, 30, 60, and 120 min, blood was sampled from the tail vein and analyzed by a handheld glucometer [121]. The analysis of the Area Under the Curve for each glucose curve was performed by subtracting the basal glucose level (t = 0 min). 

### 7.3. DEXA 

To assess body composition, DEXA scans were performed (GE Lunar PIXImus X-ray densitometer) on a subset of male and female adult (PND182)) F1 offspring mice as described previously [136]. In brief, each mouse was anesthetized throughout the duration of the scan (∼5 min) using isoflurane. Body fat, lean mass, bone mineral content, and bone mineral density were measured. 

### 7.4. Statistical Analysis 

The significance of differences among multiple groups and between 2 groups was examined using ANOVA multiple comparison tests and the Student’s *t* test, respectively. Repetitive values were analyzed using multiple *t*-test. All values are presented as means ± SEM. A *p* value ≤ 0.05 was considered significant (*). All data were analyzed using Prism data analysis software.

## Figures and Tables

**Figure 1 genes-12-01153-f001:**
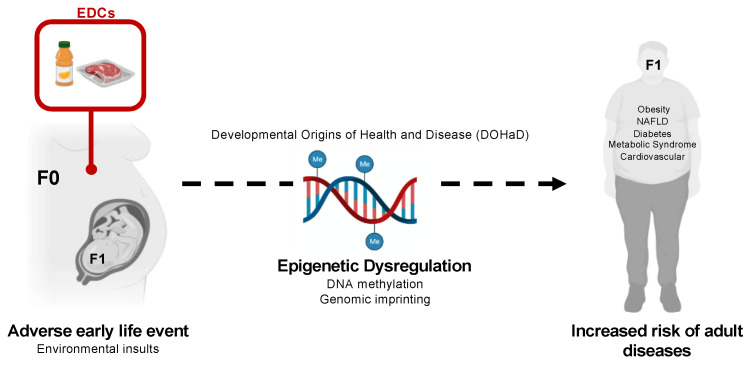
Endocrine disrupting chemicals, developmental programming, and human health. The predominant source of Environment and Development of Children (EDC) exposure is through our diet. EDCs can leach into the food by changes in temperature or pH when using plastic food containers. Human exposure to EDCs begins as early as in the mother’s womb (F0), where EDCs have been demonstrated to cross the placenta and reach the fetus. The Developmental Origins of Health and Diseases hypothesis suggests that environmental insults, including EDCs, during critical periods of early development and growth predisposes offspring (F1) to an increased risk of adult diseases. EDCs can have a profound impact on the fetal-maternal endocrine profile leading to altered fetal growth and metabolism, which increases the risk of metabolic diseases manifested during adulthood. These metabolic diseases include obesity, diabetes, metabolic syndrome, non-alcoholic fatty liver disease (NAFLD), and cardiovascular dysfunction. Although the biological mechanisms behind these associations remain unclear, an involvement of epigenetic dysregulation has been proposed to have a role in gene-environment interactions and disease risk. EDC-induced changes in metabolic gene profiles may arise from altered global or gene-specific DNA methylation patterns and consequently cause detrimental effects on genomic imprinting. (Me = Methyl group).

**Figure 2 genes-12-01153-f002:**
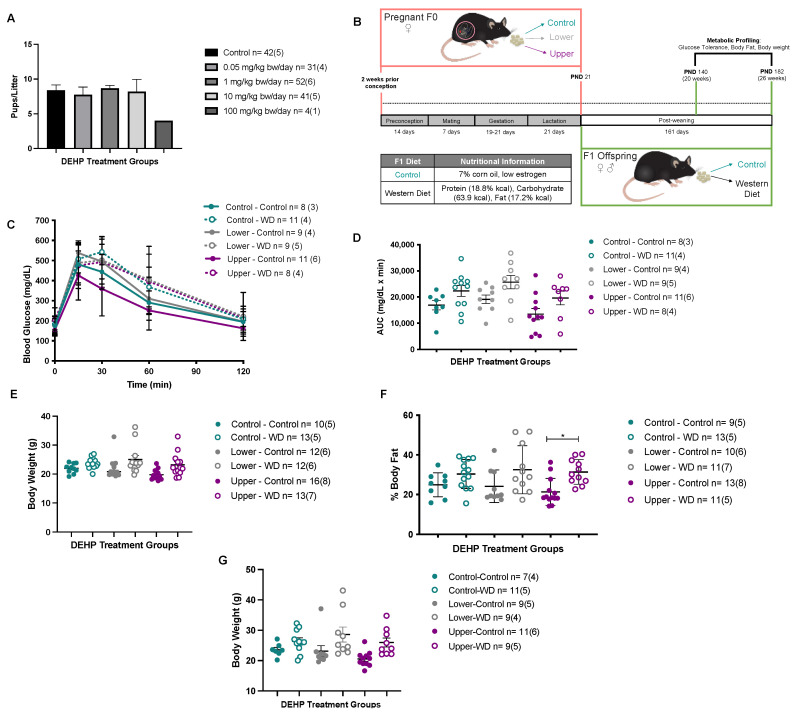
(Di-2-ethyhexyl-phthalate (DEHP) dose-response and F1 female adult offspring metabolic phenotyping profile. (**A**) Dose-response relationship between DEHP exposure in utero and E17.5 embryos per litter. (**B**) DEHP F0 exposure paradigm where F0 females were exposed to two DEHP doses (Lower: 50 ug/kg/day, Upper: 10 mg/kg/day) starting 2 weeks prior to conception until weaning. After weaning F1 offspring was placed on either a control or a Wester Diet challenge until PND182. At PND140: (**C**) Glucose tolerance test, (**D**) Glucose Area Under the Curve (AUC, mg/dL × min), and (**E**) Body weight. At PND182: (**F**) Body composition by DEXA scan and (**G**) Body weight. *N* = # of individuals (# of litters), * *p* < 0.05.

**Figure 3 genes-12-01153-f003:**
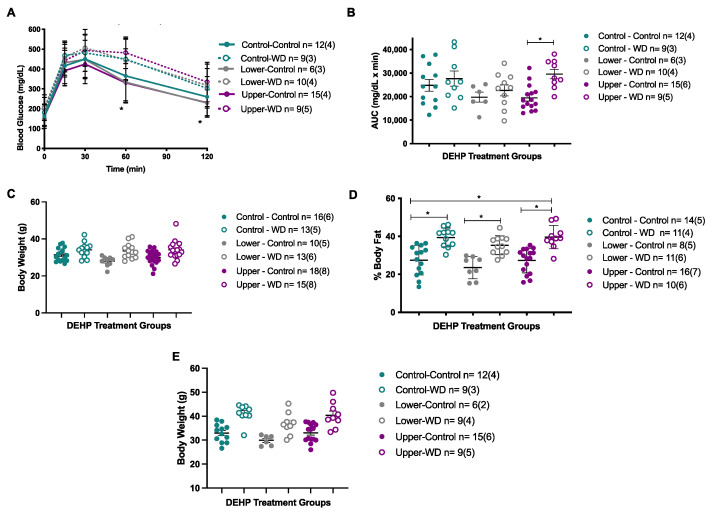
F1 male adult offspring metabolic phenotyping profile. At PND140: (**A**) Glucose tolerance test, (**B**) Glucose Area Under the Curve (AUC, mg/dL × min), and (**C**) Body weight. At PND182: (**D**) Body composition by DEXA scan and (**E**) Body weight. *N =* # of individuals (# of litters), * *p* < 0.05.

## Data Availability

The data that support the findings of this study are available from the corresponding author, MSB, upon reasonable request.

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
