# Peer review of "Environmental Exposure to Endocrine Disrupting Chemicals Influences Genomic Imprinting, Growth, and Metabolism"

_genes, 2021, doi:10.3390/genes12081153_

Round 1

Reviewer 1 Report

In this review, the authors summarize previous studies on the effects of endocrine disruptors on genomic imprinting and growth. The authors first provide an overview of genomic imprinting. Then, the authors carefully summarized the impact of Bisphenol A, Phthalates, pesticides, and Dioxins on imprinted gene expression and DNA methylation using laboratory animals such as mice and rats. Then, epidemiological studies in humans are described. Subsequently, the effects of EDGs on metabolic diseases, especially the transgenerational effects, are summarized. There are many papers that contradict each other in these research areas, making it difficult to understand comprehensively. The authors point out these problems as well, so this manuscript is an extremely useful review.

Furthermore, this manuscript contains the authors' own data on the effects of DEHP exposure on the next generation at the later part of this manuscript. Although the experimental data showed no statistically significant differences, this data is very important because it emphasizes the difficulties of research in this area. The "Recommendations for Future Research on EDCs and Environmental Epigenetics" at the end of this manuscript is also quite appropriate and important and should be kept in mind by all researchers in this area. Overall, this manuscript is very well written and will be read and cited by many researchers once published, so I strongly recommend that it be accepted for publication after a minor revision.

Minor points

  1. It would be better to describe H3K27me3-mediated non-canonical imprinting, which is recently reported, as it is critical for placental development.
  2. Tables 1 and 2 may be easier to read if laid out in a landscape format rather than a portrait format.
  3. Figure 2 and 3: Adding the illustration of the experimental design would be helpful for readers to understand the results.
  4. Figure 2 and 3: What does AUC indicate? Is it AUC of detectable DEHP in the blood? What are the units for the Y-axis in Figures 2C and 3C?

Reviewer 2 Report

Robles-Matos et al present a comprehensive and timely review of the impacts of EDCs on genomic imprinting, growth and metabolism, and introduce a novel model of developmental phthalate exposure. The review will be of broad interest to researchers in the fields of developmental programming, epigenetics, metabolism and toxicology.

This review is exceptional in a number of ways. First, it is very comprehensive, providing a complete picture of relevant rodent and human studies. The tables are well-presented and highly informative. Second, the review draws comparisons between studies, identifying consistencies and inconsistencies, and providing potential explanations for any differences observed. Third, the review is fair in assessing the limitations of both human and animal studies that have been performed previously, including studies from the authors’ own lab. Fourth, the authors introduce their preliminary studies using a novel mouse model of developmental phthalate exposure and integrate these findings with previously published data. Finally, they present a bulleted list of thoughtful recommendations for future research which will provide a foundation for the developmental of new experiments and grant applications.

I have not previously seen such a well-written review as a first submission and the authors should be commended on their efforts. I believe this is highly worthy of publication and will be widely-read and cited.

I have one request and two suggestions:

Request: in the methods section, please elaborate on the multiple comparisons test used for data analysis. The methods state that an ANOVA was used but which post-hoc test was used to identify the groups between which differences exist?

Suggestions:

  1. The methyl marks in figure 1 look more like histone post-translational modifications than cytosine methylation since they originate from the nucleosomes. I would consider modifying this for clarity.
  2. There is not much discussion of potential mechanisms for how EDCs may modulate DNA methylation at ICRs (and elsewhere in the genome). Indeed, not a great deal is known but it might be worth stating this and suggesting it as a recommendation for future research.
